# E²GAN: Efficient Training of Efficient GANs for Image-to-Image Translation

## Abstract

One highly promising direction for enabling flexible *real-time on-device* image editing is utilizing data distillation by leveraging large-scale text-to-image diffusion models, such as Stable Diffusion, to generate paired datasets used for training generative adversarial networks (GANs). This approach notably alleviates the stringent requirements typically imposed by high-end commercial GPUs for performing image editing with diffusion models. However, unlike text-to-image diffusion models, each distilled GAN is specialized for a specific image editing task, necessitating costly training efforts to obtain models for various concepts. In this work, we introduce and address a novel research direction: *can the process of distilling GANs from diffusion models be made significantly more efficient?* To achieve this goal, we propose a series of innovative techniques. First, we develop an attention-based network architecture tailored for efficient image-to-image translation on mobile devices, which yields faster inference speeds, reduces the number of parameters, and lowers computational costs compared to existing image-to-image models. Second, we introduce a hybrid training pipeline that efficiently adapts a pre-trained text-conditioned GAN model to different concepts while substantially reducing computational costs. Moreover, this approach significantly minimizes the storage requirements for each concept. Third, we investigate the minimal amount of data necessary to train each GAN, further reducing the overall training time. Extensive experiments demonstrate that we can efficiently empower GANs with the ability to perform real-time high-quality image editing on mobile devices with remarkable reduced training cost and storage for each concept.

## 1 Introduction

Recent development of diffusion-based image editing models has witnessed remarkable progress in synthesizing contents containing photo-realistic details with full of imagination (Saharia et al., 2022; Rombach et al., 2022; Ramesh et al., 2021; 2022). Albeit being creative and powerful, these generative models typically require a huge amount of computation even for inference and storage for saving weights. For example, Stable Diffusion (Rombach et al., 2022) has more than one billion parameters and takes 30 seconds to conduct iterative denoising process to get one image on T4 GPU. Such low-efficiency issue prohibits their real-time application on mobile devices (Li et al., 2023).

Exiting works try to tackle the problem through two main directions. One is accelerating the diffusion models by designing efficient model architecture or reducing the number of denoising steps (Salimans & Ho, 2022; Meng et al., 2022; Li et al., 2022a; Kim et al., 2023). However, these efforts still struggle to obtain models that can run in real-time on mobile devices (Li et al., 2023). Another area focuses on data distillation, where diffusion models are leveraged to create datasets to train other mobile-friendly models, such as generative adversarial networks (GANs) for image-to-image translation (Zhao et al., 2021; Parmar et al., 2023). Nevertheless, although GAN is efficient for on-device deployment, each new concept still asks for the *costly training* of a GAN model from *scratch*.

In this work, we propose and aim to address a new research direction: *can the GAN models be trained efficiently under the data distillation pipeline to perform real-time on-device image editing?*

To tackle the challenge, we introduce **E²GAN**, powered with the following techniques for the **E**fficient training and **E**fficient inference of **GAN** models with the help of diffusion models:

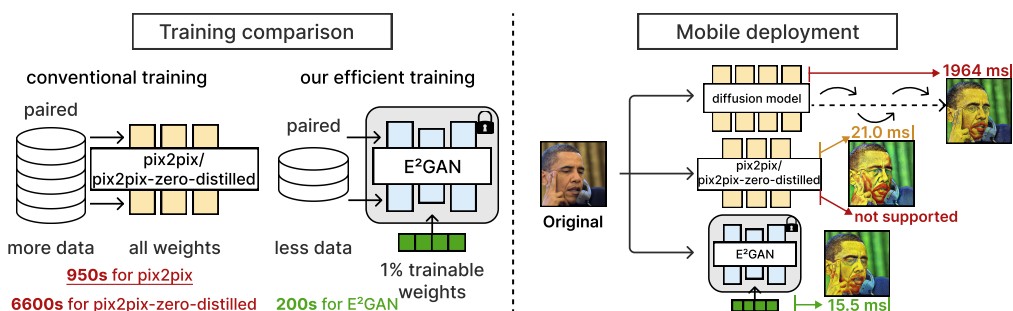

Figure 1: **Overview of E$^2$GAN.** *Left: Training Comparison.* Conventional GAN training, such as pix2pix (Isola et al., 2017) and pix2pix-zero-distilled that distills Co-Mod-GAN (Zhao et al., 2021) using data from pix2pix-zero (Parmar et al., 2023), requires all the weights trained from scratch, while our efficient training significant reduces the training cost by only fine-tuning $1\%$ weights with only *portion* of training data. **Right**: *Mobile Inference Comparison.* Our efficient on-device model can achieve real-time (30FPS, iPhone 14) runtime and is faster than pix2pix and diffusion model, while the pix2pix-zero-distilled model (Co-Mod-GAN) is not supported on device.

- First, we develop an efficient image-to-image translation architecture with attention mechanism. Different from conventional image-to-image models that are built solely with convolution (CONV) operations, we design the efficient transformer block for our model to achieve fast inference speed with fewer parameters on mobile devices (as in Fig. 1 *Right*), while maintaining high-performance. This model further enables us to smoothly incorporate text condition that can help efficiently adapt a pre-trained model into different tasks.

- Second, we substantially improve the training and storage efficiency of our image-to-image model in two ways. **(1)** We improve the GAN model into a text conditional image generation model, named as base GAN model, and train it with various prompts and the corresponded edited images obtained from diffusion models. For the new editing concepts, we can then fine-tune the base GAN model, rather than training models from scratch, to reduce the training cost. **(2)** We identify that instead of training all the weights, only partial weights of the base GAN model are necessary to be fine-tuned for new concepts (as in Fig. 1 *Left*), bringing two advantages – both the training cost and saving storage for each new concept are significantly reduced.

- Third, we investigate the amount of data for fine-tuning the base model for various concepts. The reducing the amount of training data helps reduce the training cost and time for adapting the base model to new concepts.

We show extensive experiments results to demonstrate that by using our approach, we can efficiently distill the image editing capability from a large-scale text-to-image diffusion model into GAN models via data distillation (examples in Fig. 5). The distilled GAN model showcases real-time image editing capabilities on mobile devices. We hope our work can shed light on how to democratize the diffusion models into efficient on-device computing.

## 2 RELATED WORKS

**Generative Models** learn the joint data distribution to generate new samples, such as VAEs (Kingma & Welling, 2013; Rezende et al., 2014), GANs (Goodfellow et al., 2020; Zhu et al., 2017; Park et al., 2019), auto-regressive models (Van Den Oord et al., 2016; Salimans et al., 2017; Van Den Oord et al., 2016; Menick & Kalchbrenner, 2018; Yu et al., 2022), and diffusion models (Sohl-Dickstein et al., 2015; Ho et al., 2020; Nichol & Dhariwal, 2021; Song et al., 2020a;b; Dhariwal & Nichol, 2021). Among these, diffusion models demonstrate strong capability of generating images with high-fidelity (Ramesh et al., 2022; Rombach et al., 2022), at the cost of bulky model size and numerous sampling steps during inference. Several studies try to accelerate the image generation process of the diffusion models (Salimans & Ho, 2022; Meng et al., 2022; Li et al., 2022a). However, they still struggle to achieve real-time on-device generation. On the contrary, GANs are more efficient in terms of model size and inference speed for image editing (Li et al., 2020; Jin et al., 2021; Wang et al.,

2020). To this end, we leverage the approach of data distillation to transfer knowledge from diffusion models to lightweight GANs that are compatible with real-time inference on mobile devices.

**Efficient GANs.** Exiting works actively explore the reduction of the inference runtime for GANs by using various model compression techniques, such as efficient architecture design (Li et al., 2020; Jin et al., 2021), network pruning and quantization (Wang et al., 2020; 2019), and neural architecture search (Wang et al., 2020; Fu et al., 2020). On the other hand, the research about training cost savings for GANs is quite limited, as most works typically train all the parameters of a GAN model from scratch for the image-to-image translation task, involving large computing efforts. This work aims to fine-tune a very small portion, *i.e.,* $1\%$, of the pre-trained models with the partial training data to reduce the training cost. Thus, the training of GAN can be tiny in terms of both parameters and data. There are many efforts on efficient training (Huang et al., 2019; Köster et al., 2017), in particular the sparse training (Evci et al., 2020; Lee et al., 2019; Yuan et al., 2022). However, these methods rely on the mask of trainable parameters, which in turn is determined during training with a huge bunch of data. In contrast, our method adopts pre-defined learnable components, and only fine-tunes on a small fragment of data to make the transfer learning progress efficient and effective.

## 3  METHODS

Here we present our approach that efficiently transfers the image editing capability from large-scale text-to-image diffusion models to on-device real-time GANs. In the following sections, we first give an overview of our knowledge transfer pipeline (Sec. 3.1). Second, we introduce our efficient image-to-image model that has *faster inference speed, higher performance,* and *less number of parameters* than conventional CONV-based models (Isola et al., 2017; Zhu et al., 2017) (Sec. 3.2). Third, we study efficient training strategies to get a series of on-device models with *fewer* training costs and *less* storage, while maintaining high-quality image generation (Sec. 3.3).

### 3.1  OVERVIEW OF KNOWLEDGE TRANSFER PIPELINE

**Benefits of Knowledge Transfer.** Despite the powerful image generating capability of text-to-image diffusion models, they suffer from large model size and inefficient sampling (Li et al., 2023). For example, the denoising UNet of Stable Diffusion (Rombach et al., 2022) has 860 millions of parameters. Even the latest efficient diffusion model, SnapFusion (Li et al., 2023), costs around 2 seconds to generate a single image on iPhone 14 Pro, limiting the real-time applications for diffusion models. On the contrary, the image-to-image GANs are typically light-weight with much fewer parameters that can synthesize one image with only one forward pass. For instance, a pix2pix model with ResNet-based network has a latency of around 21ms on iPhone 14 Pro. Thus, these efficient architectures are more compatible with the mobile devices to achieve real-time image generation. To leverage the benefits of both models, we employ GANs as the image editing model and transfer knowledge of diffusion models to GAN by training the GANs with data created by the diffusion model. Such a data distillation pipeline serves as the foundation for our whole framework.

**Pipeline for Dataset Creation.** To enable the data distillation, we use the diffusion models to edit real images to obtain the edited images, forming pairs of data along with the used text prompts to create the training datasets, which then be utilized to train the image-to-image GAN model. The real images come from FFHQ (Karras et al., 2019) and Flickr-Scenery (Cheng et al., 2022), covering diverse content and are challenging for content editing. For diffusion models, we choose the recent works for image editing, such as Stable Diffusion (Rombach et al., 2022), Instruct-Pix2Pix (IP2P) (Brooks et al., 2022), Null-text Inversion (NI) (Mokady et al., 2022), ControlNet (Zhang & Agrawala, 2023), and InstructDiffusion (Geng et al., 2023).

**Training Objectives.** With paired images and the associated prompts, we train the efficient GANs for image translation by using the conventional adversarial loss. Specifically, given the original image $\mathbf{x}$ and the editing prompt $\mathbf{c}$, the image generator $\mathcal{G}$ and discriminator $\mathcal{D}$ are jointly optimized as follows:

$$\min_{\theta_g} \max_{\theta_d} \underbrace{\mathbb{E}_{\mathbf{x},\tilde{\mathbf{x}}^c}\left[\log \mathcal{D}(\mathbf{x}, \tilde{\mathbf{x}}^c; \theta_d)\right] + \mathbb{E}_{\mathbf{x},\mathbf{z},\mathbf{c}}\left[\log(1 - \mathcal{D}(\mathbf{x}, \mathcal{G}(\mathbf{x}, \mathbf{z}, \mathbf{c}; \theta_g); \theta_d))\right]}_{\text{conditional GAN loss}} + \lambda \underbrace{\mathbb{E}_{\mathbf{x},\tilde{\mathbf{x}}^c,\mathbf{z}}\left[\|\tilde{\mathbf{x}}^c - \mathcal{G}(\mathbf{x}, \mathbf{z}, \mathbf{c}; \theta_g)\|_1\right]}_{\ell_1 \text{ loss}}, \quad (1)$$

where $\tilde{\mathbf{x}}^c$ denotes the image generated by the diffusion model $\epsilon$ conditioned on the concept $\mathbf{c}$ of the target style, *i.e.*, $\tilde{\mathbf{x}}^c = \epsilon(\mathbf{x}, \mathbf{c})$, $\mathcal{G}$ and $\mathcal{D}$ denote the generator and discriminator function parameterized

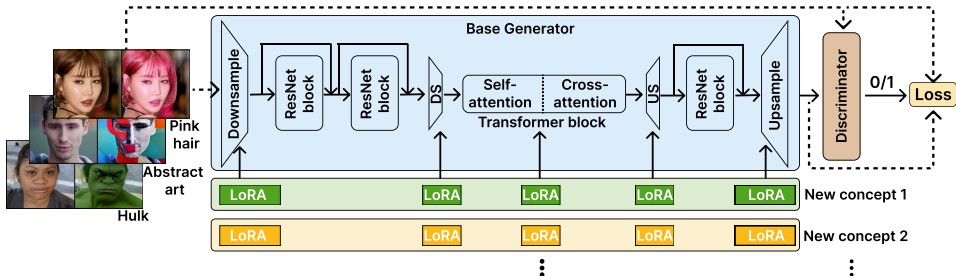

Figure 2: **Overview of the model architecture for E²GAN.** The generator is composed of down-/up-sampling layers, 3 RBs, and 1 TB. The base generator is trained on multiple representative concepts. New concepts are achieved by only fine-tuning LoRA parameters on crucial layers.

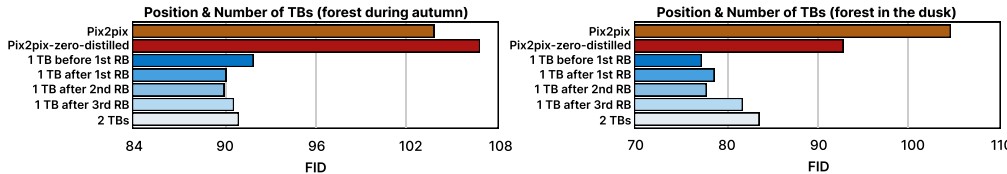

Figure 3: **FID comparison** of applying TBs in image generators trained on two datasets (*Left:* `forest during autumn`, *Right:* `forest in the dusk`). Vertical axis shows the position to inserting TBs. Pix2pix-zero-distilled uses pix2pix-zero for creating datasets to train Co-Mod-GAN (Ramesh et al., 2021).

by $\theta_g$ and $\theta_d$, respectively, $\mathbf{z}$ is a random noise introduced to increase the stochasticity of output, and $\lambda$ can be used to adjust the relative importance between the conditional GAN loss and $\ell_1$ loss.

## 3.2 NETWORK ARCHITECTURE OF EFFICIENT IMAGE-TO-IMAGE MODEL

Here we elaborate the details on the construction of the image-to-image model $\mathcal{G}$ (overall architecture in Fig. 2). We intend to make the architecture of $\mathcal{G}$ being effective yet as simple as possible. Existing designs for $\mathcal{G}$ are mainly composed of convolutional (CONV) layers, *e.g.*, 9 ResNet Blocks (RBs) (Isola et al., 2017; Zhu et al., 2017; Park et al., 2020). On the other hand, transformer blocks (TBs) with multi-head self-attention (Dosovitskiy et al., 2021) have demonstrated huge success on image classification tasks (Liu et al., 2021; Li et al., 2022b) and diffusion models (Rombach et al., 2022; Brooks et al., 2022), while applying them in GANs are less studied. We investigate the number and position of TBs in GANs for the image translation task. The TBs are incorporated into a classic ResNet generator, where we reduce the number of RBs from 9 to 3 to save the computation cost.

**Number of TBs.** We train models with different architecture designs, *e.g.,* different number of TBs, and calculate the FID (Heusel et al., 2017) between the images generated by GANs and diffusion models (results shown in Fig. 3). Interestingly, we find that one TB is enough for generating high quality images. Introducing more TBs does not further improve the performance yet brings in more computation cost. Notice that to reduce the inference cost of the introduced TB, we apply a downampling operation to halve the feature map size before sending it into the TB, and use an upsampling layer to recover the feature map size for the following operations.

**Position of TBs.** Additionally, we find that the position of the TB is important for the final performance of the image generation. First, the TB should be placed between the last downsample layers and the first upsample layers to avoid high computations on mobile devices, due to the high resolution of features. Second, we apply attention to different positions of the network bottleneck. Particularly, the TB can be inserted between one of the following: (1) before the first RB; (2) after the first RB; (3) after the second RB; and (4) after the third RB. As evident in Fig. 3, all these options lead to a generator with better performance than the conventional CONV-only networks used in pix2pix (Isola et al., 2017) and pix2pix-zero-distilled (Parmar et al., 2023). For our model, we place the TB after the second RB. With the introduction of self-attention in our image translation model, we reduce the number of residual blocks and obtain the model with fewer parameters, less computation cost, and better performance (more results in Tab. 1).

### 3.3 EFFICIENT TRAINING OF GAN MODELS

Diffusion-based generative models can perform image editing on-the-fly, while existing light-weight GAN-based networks require training to be adapted to the new concept. The training of GAN models for various image translation tasks require substantial computation costs. Additionally, there is high storage demand for saving the trained weights. To mitigate such training and storage costs, we introduce three main techniques to reduce the number of trainable parameters and the demanded data for model fine-tuning: *First*, we establish a *base GAN model* equipped with generalized features and representations, ready to be leveraged for new concepts (Sec. 3.3.1). *Second*, starting from the base model, we identify key parameters to optimize during fine-tuning for a new concept, bolstered by the application of Low-Rank Adaptation (LoRA) (Hu et al., 2021) to further reduce the number of parameters (Sec. 3.3.2). *Third*, we explore the possibility of tiny fine-tuning where the training data are first clustered and only those near the cluster centers are used (Sec. 3.3.3).

#### 3.3.1 BASE GAN MODEL CONSTRUCTION

To obtain model weights for a new target concept with as few training efforts as possible, we explore transfer learning from a pre-trained base GAN model, instead of training from scratch. The base model should possess the capability of a more general features and representations, which can be learned from multiple image translation tasks, allowing the new concept to leverage existing knowledge.

Thus, we opt to train the base model on a mixed datasets comprising diverse concepts.

Table 1: **Comparison of model size, FLOPs, and latency** for different works (Li et al., 2023; Isola et al., 2017; Parmar et al., 2023). Co-Mod-GAN (Zhao et al., 2021) is trained following the pipeline in Pix2pix-zero (Parmar et al., 2023). Reported latency is averaged over 100 runs on iPhone 14 Pro.

| Model | Param num | FLOPs | Latency |
|---|---|---|---|
| SnapFusion | 861M | >1T | 1956 ms |
| Pix2pix with 9 RB | 11.4M | 56.9G | 21.0 ms |
| Co-Mod-GAN | 79.2M | 98.2G | Not supported |
| **3RB+1TB** | **7.1M** | **23.6G** | **15.5 ms** |

To enable the base model to learn multiple concepts, we introduce an additional condition involving *cross-attention* between the text information of the concept and image feature maps by taking advantage of our efficient model architecture with TB (as the bottleneck in Fig. 3). With the cross-attention module, our architecture is finalized with the model size, FLOPs, and latency benchmark in Tab. 1. The base model is trained on a subset of concepts denoted as $\mathcal{C} = \{\mathbf{c}_1, \cdots, \mathbf{c}_K\}$, where each concept $\mathbf{c}_k$ is selected among different concepts by K-means clustering (Lloyd, 1982) based on the average of the CLIP image embedding (Radford et al., 2021) of uniformly sampled images.

#### 3.3.2 CRUCIAL WEIGHTS FOR FINE-TUNING

To save the training and storage cost, we reduce the number of trainable parameters during fine-tuning. Specifically, we pre-define trainable layers that occupy a small portion of weights from the base model. Then, we apply LoRA on top of the trainable layers. In this way, we only optimize 1.29% of the weights from the base model during the fine-tuning, greatly reducing the training costs.

Inspired by the recent work of customized diffusion (Kumari et al., 2022), where a pre-trained diffusion model can be fine-tuned to a personalized version with updating only partial weights, we explore the feasibility to identify the minimal set of tunable weights for GAN. We aim to determine such a set that is sufficient for fine-tuning the base model to a new target concept. To this end, we analyze the components of the GAN model, which

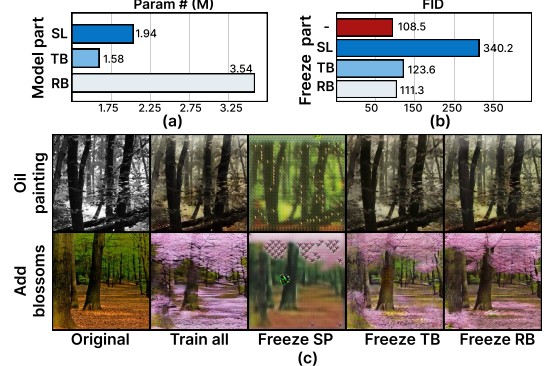

Figure 4: **Crucial weights analysis** via freezing partial weights in the base model. **(a)** Number of parameters for each part of the base model; **(b)** Averaged FID across 10 different concepts on the Flicker-Scenery dataset when freezing partial weights of base model. '-' indicates fine-tune all the weights; **(c)** The generated images when freezing each part of the base model.

mainly consist of three parts: (1) sampling layers (SL) with downsampling and upsampling; (2) transformer block (TB); and (3) intermediate ResBlock (RB).

**Identifying Crucial Layers.** We systematically and empirically study the impact of each part in the image-to-image task by freezing each part individually, with results provided in Fig. 4. Combining Fig. 4(b) & (c), we see that SL plays a more crucial role in maintaining the quality of generated images, identified by the high FID score and low image quality, than other parts. SL might be more crucial for constructing the desired output texture, yet intermediate RB might contain lower-level information that are common among styles. Meanwhile, compared to RB, TB has a fewer amount of parameters (1.58M *v.s.* 3.54M in Fig. 4(a)), while it is more important in keeping performance (123.6 *v.s.* 111.3 in Fig. 4(b)). Considering the situation with a limited training budget, RB has a lower priority to be optimized.

**LoRA on Crucial Layers.** From the perspective of maintaining image generating quality, it is better to include TB in training as self-attention modifies the image with a better holistic understanding and the cross-attention module takes the information from the given target concept. Combing SL and TB leads to 3.42M parameters to be updated, taking account $47.90\%$ of the entire model weights. To fine-tune the crucial layers with much fewer trainable parameters, we investigate the best way of incorporating Low-Rank Adaptation (LoRA) (Hu et al., 2021) into GAN training, which introduces two trainable low-rank weight matrices besides the original weight for each layer identified as crucial.

**Rank for LoRA.** During fine-tuning, the weights of base model are *frozen*, while the two low-rank matrices with much fewer parameters are updated to save computation and storage costs. For instance, for a CONV layer $i$ with weights $\theta_i \in \mathbb{R}^{h \times w \times k_h \times k_w}$, we apply two low-rank matrices with rank $r_i$, *i.e.,* $\theta_i^A \in \mathbb{R}^{h \times r_i \times k_h \times k_w}$ and $\theta_i^B \in \mathbb{R}^{r_i \times w \times 1 \times 1}$, to approximate the gradient update $\nabla \theta_i$. Given multiple crucial layers, determining the appropriate rank for *each of them* is important. Prior works mostly rely on manual setting (Hu et al., 2021) for deciding the rank value, due to a huge search space for the rank. However, in our task, the rank should be pre-fixed for different concepts to avoid the rank search process when a new concept comes. To tackle this challenge, we randomly sample $K$ concepts and conduct a simple yet effective rank search process. For each concept, we start by assigning $r_i$ as 1 for each crucial layer $i$, and upscale the rank for every $e$ epochs by doubling the rank value, until $r_i$ reaches the upper threshold $\tau_i$ for the layer $i$. The threshold $\tau_i$ is determined by the size of the weight. We evaluate the FID performance at the end of each $e$ training epochs. If the performance saturates, the rank value from the best FID performance setting is returned as the rank for the concept. Typically, a larger rank can provide more model capability. Thus, the largest returned rank among the $K$ selected concepts is viewed as $r^*$ for the future use of a new concept.

### 3.3.3 Similarity Clustering (SC) for Training Data Reduction

Reducing the amount of training data can directly result in a reduction in the training time. Thus, we aim to investigate data efficiency as a means of decreasing the training workload in addition to the curcial weight update for E$^2$GAN. We find not all data are indispensable for reliable training, but only a small subset is necessary. We obtain this small subset in an unsupervised manner with a selection of the data crowding around the clustering center on the whole dataset.

To identify the small subset of essential data, we conduct unsupervised learning to analyze the structure of the training data. We first extract an embedding for each image $\mathbf{x}$ with an extractor $\mathcal{E}$. Then, we apply clustering on the embeddings by the K-Means algorithm (Lloyd, 1982) to obtain $K < N$ clusters ($N$ it the total number of images), each with center $\mu_k$. The embeddings within the same cluster have a closer distance among each other, indicating a higher *similarity* of the data points. To reduce the data amount while maintaining data diversity for the good generalization ability of the model, one data point, which is the closet to the center $\mu_k$, is selected for each of the $K$ clusters.

With our data selection method using $K$ clusters, we further reduce the number of training iterations by $N/K$ times. In contrast to prior methods involving additional computations in the training process to shrink the dataset (Yuan et al., 2021; Wang et al., 2022), our SC data reduction is tailored for expediting the training of image editing tasks. It reduces the training data volume directly before the training process without incurring any additional costs during the training.

## 4 Experiments

In this section, we provide the detailed experimental settings and results of our proposed method. More details as well as some ablation studies can be found in the Appendix.

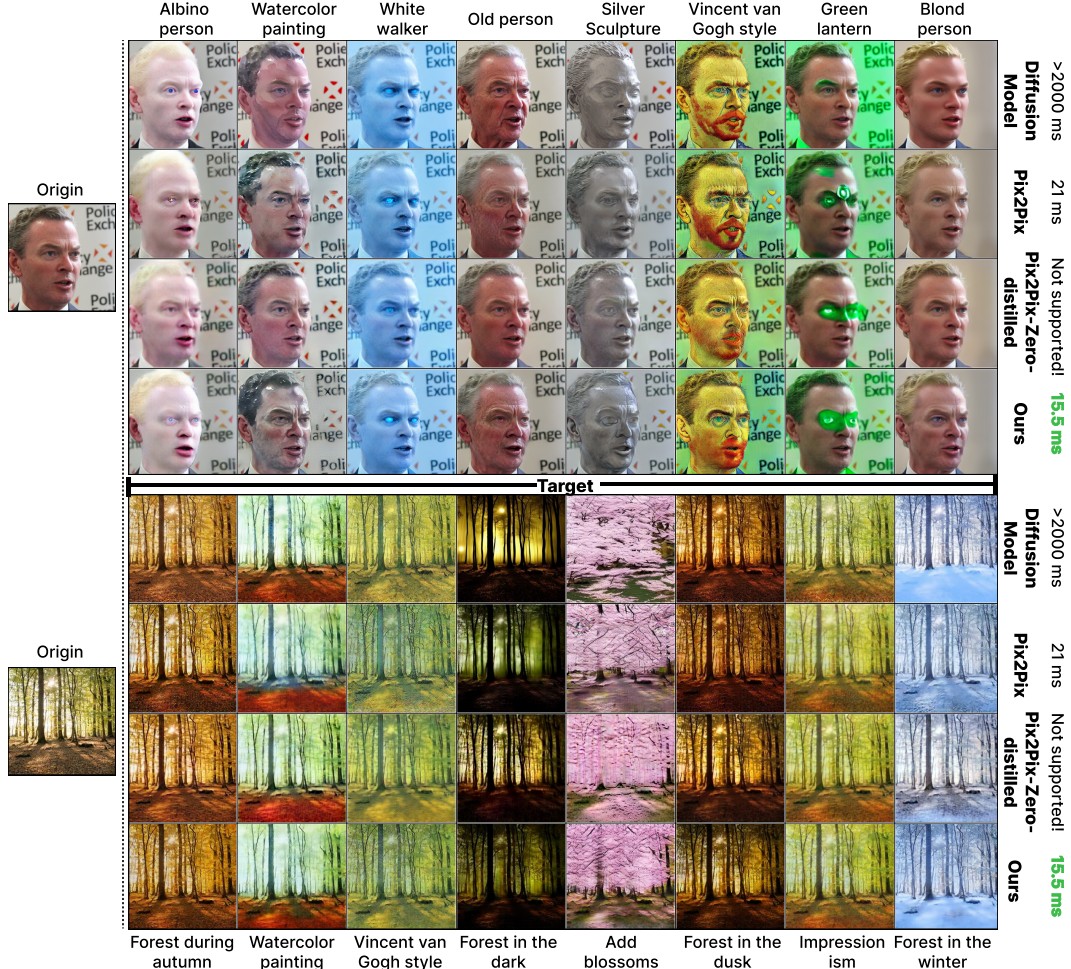

Figure 5: **Qualitative comparisons** on various tasks. The *leftmost* column shows two original images and the remaining columns present the corresponding synthesized images in the target concept domain, where target prompts are shown at the bottom row. We provide images generated by various models.

**Paired Data Preparation.** We verify our method on $1,000$ images from FFHQ dataset (Karras et al., 2019) and Flickr-Scenery dataset (Cheng et al., 2022) with image resolution as $256 \times 256$. The images in the target domain are generated with several different text-to-image diffusion models, including Stable Diffusion (Rombach et al., 2022), IP2P (Brooks et al., 2022), NI (Mokady et al., 2022), ControlNet (Zhang & Agrawala, 2023), and InstructDiffusion (Geng et al., 2023). The generated images with the best perceptual quality among diffusion models are selected to form with the real images into paired datasets. To perform training and evaluation of GAN models, we divide the image pairs from each target concept into training/validation/test subsets with the ratio as $80\%/10\%/10\%$.

**Baselines.** We compare $E^2GAN$ with image-to-image translation methods like pix2pix (Isola et al., 2017) (image generator with 9 ResNet blocks) and pix2pix-zero-distilled that distills Co-Mod-GAN (Zhao et al., 2021) using data generated by pix2pix-zero (Parmar et al., 2023).

**Training Setting.** We follow the standard approach that alternatively updates the generator and discriminator (Goodfellow et al., 2020). The training is conducted from an initial learning rate $2e-4$ with mini-batch SGD using Adam solver (Kingma & Ba, 2014). The total training epochs is set to 100 for $E^2GAN$, and 200 for pix2pix (Isola et al., 2017) and pix2pix-zero-distilled (Parmar et al., 2023) for them to converge well. For SC (Sec. 3.3.3), we choose the cluster number as 400 and use the feature extractor $\mathcal{E}$ as FaceNet (Schroff et al., 2015) on FFHQ dataset and CLIP image encoder (Radford et al., 2021) on Flicker Scenery dataset. To train the base model, we use 20 prepared tasks/datasets from FFHQ dataset and 7 from Flicker Scenery dataset.

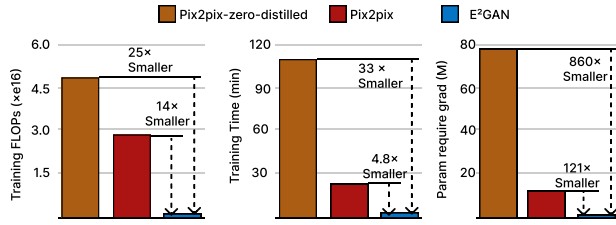

Figure 6: **Training cost comparison** of conventional GAN and E$^2$GAN. **Left**: Training FLOPs. **Middle**: Training time. **Right**: Number of parameters that required gradient update, which also equals to the weights need to be saved for a concept.

Table 2: **FID comparison.** FID is calculated between the images generated by GAN-based approaches and diffusion models. Reported FID is averaged across different concepts (30 for FFHQ and 10 for Flicker Scenery).

| Dataset
Method | FFHQ | Landscape |
|---|---|---|
| Pix2pix | 86.03 | 114.2 |
| Pix2pix-zero-distilled | 87.76 | 132.6 |
| **E$^2$GAN** | **80.28** | **109.37** |

**Evaluation Metric.** We compare the images generated by GAN-based models and diffusion-based models by calculating Clean FID proposed by Parmar et al. (2022) on the test sets.

## 4.1 EXPERIMENTAL RESULTS

**Qualitative Results.** The synthesized images in the target domain obtained by E$^2$GAN and other methods are shown in Fig. 5. The original images are listed at the leftmost column, and the synthesized images for the target concept obtained by diffusion models, pix2pix, pix2pix-zero-distilled, and E$^2$GAN are shown from top to bottom. The tasks span a wide range, such as changing the age, artistic styles, and editing the seasons. According to the results, E$^2$GAN is able to modify the original images to the target concept domain by updating only the LoRA parameters. For instance, for the `green lantern` prompt on FFHQ dataset, diffusion model fails to modify the image, pix2pix and pix2pix-zero-distilled add colors to wrong areas, while E$^2$GAN generates the image that fits the prompt well. As for the `add blossoms` prompt on the Flicker Scenery dataset, E$^2$GAN preserves the structure of the original image better than other models while editing the image as desired.

**Quantitative Comparisons** between E$^2$GAN and other baseline methods are provided in Tab. 2. Note that for each concept, pix2pix and pix2pix-zero-distilled are trained on the whole training dataset of 800 samples. E$^2$GAN begins with a base model and is fine-tuned with only 400 samples to obtain models for different target concepts. The results demonstrate E$^2$GAN is able to reach a even better FID performance than the conventional GAN training techniques like pix2pix and pix2pix-zero-distilled, indicating high-fidelity of generated images.

**Training Cost Analysis** between E$^2$GAN and other approaches are provided in Fig. 6. Compared with pix2pix and pix2pix-zero-distilled, E$^2$GAN greatly saves the training FLOPs of $14\times$ and $25\times$, respectively, and accelerates the training time by $4.8\times$ and $33\times$, respectively. Moreover, E$^2$GAN only requires updating 0.092M parameters for a new concept, greatly saving the storage requirement when training models for various tasks/concepts, *i.e.,* $869\times$ less than pix2pix-zero-distilled.

Notably, E$^2$GAN requires *much fewer* trainable parameters, training data, and training time than other GAN-based approaches to reach even *better* generation quality, *i.e.,* E$^2$GAN has lower FID than pix2pix on FFHQ (80.28 *v.s.* 86.03). Furthermore, E$^2$GAN enjoys a faster inference speed on mobile devices (Tab. 1). The good performance of E$^2$GAN is originated from our effective framework design, including the efficient model architecture (Sec. 3.2) and efficient training strategy (Sec. 3.3).

## 4.2 ABLATION ANALYSIS

We provide ablation analysis to understand the impact of each components. We first study the effectiveness of the base model determination. After that, we provide an analysis on the LoRA rank search. Finally, we discuss the effect of our data selection.

**Analysis of Base Model Determination.** We study the impacts of our base model determination method discussed in Sec. 3.3.1 by comparing our method with the following three settings: (1) train the base model on 20 *random* concepts; (2) train the base model on 200 artist concepts; (3) train the

Table 3: **Analysis (FID) of various base models** on FFHQ.

| Base model
Concept | Ours | 20
random | 200 art
concepts | Single
concept |
|---|---|---|---|---|
| White walker | **40.18** | 53.92 | 40.32 | 51.99 |
| Blond person | **48.01** | 52.77 | 61.50 | 55.58 |
| Sunglasses | **38.49** | 40.54 | 41.37 | 44.12 |
| Vangogh style | 71.82 | 78.58 | **68.21** | 78.06 |

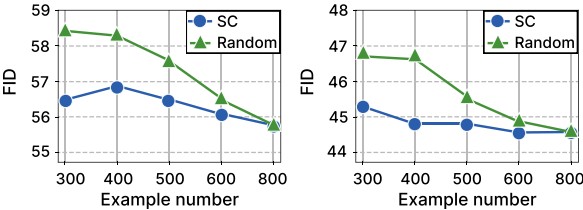

Figure 7: **Comparisons of Data Selection Rule.** Prompts for the *left* and *right* figures are `old person` and `put on a pair of sunglasses`, respectively.

Table 4: **Analysis of searching LoRA rank** on the Flicker Scenery dataset. The reported FID values are averaged over 10 different target concepts.

| Scheme | FID | # of Param |
|---|---|---|
| **Our searched** | **109.37** | 0.092M |
| Upcale 1× | 130.98 | 0.056M |
| Upcale 4× | 111.42 | 0.164M |
| Random | 129.87 | 0.100M |

base model on a *single* concept `old person` from the FFHQ dataset. The results are demonstrated in Tab. 3. Note our method is obtained by training on 20 selected representative concepts. The results indicate our base model construction outperforms or matches the alternatives across the evaluated concepts. This underscores the efficacy of our base model in enhancing performance. In contrast, the single concept base model generally performs worse. Furthermore, simply increasing the amount of concepts does not necessarily leads to better performance as indicated by training the base model with 200 art concepts.

**Analysis of LoRA on Crucial Layers.** Tab. 4 presents an evaluation of the effectiveness of our LoRA rank search on the Flicker Scenery dataset. The table reports the FID averaged across 10 different target concepts, as well as the number of LoRA parameters for various schemes. We compare our method with the other three settings: (1) upscale the rank 1× for each crucial layer by doubling the rank from the initialization until the rank reaches the threshold; (2) upscale the rank 4× for each crucial layer from the initialization; and (3) random assign ranks for the crucial layers. Our searched scheme achieves the lowest FID value of 109.37 while maintaining a relatively low number of parameters as 0.092M. This demonstrates the effectiveness of our LoRA rank search approach.

**Analysis of Cluster Number of Data Selection.** To investigate our sampling rule SC for obtaining training samples (proposed in Sec. 3.3.3 to reduce the number of training data), we compare it with randomly sampling method. Random sampling is implemented as shuffling the training data randomly and only accessing the first $K$ examples as training data. The comparisons are conducted with different number of training samples $K$. We show the results in Fig. 7 and can draw the following observations. First, SC provides better FID performance in all scenarios, indicating the effectiveness of our sampling method by enriching data diversity. Second, the cluster number, *i.e.,* the number of target training samples, influences the SC performance to some extent. More training examples (clusters) do not necessarily lead to better performance. Furthermore, SC can work for a wide range of different number of training samples by providing models with good FID performance.

## 5 CONCLUSION

This paper addresses the growing demand for efficient on-device image editing by introducing a novel research direction, that is the efficient training of efficient GAN models through distilling the large-scale text-to-image diffusion models with data distillation. The proposed framework, E²GAN, incorporates a hybrid training pipeline that can efficiently adapt a pre-trained text-conditioned GAN model, which has real-time inference speed on mobile devices, to different concepts, while significantly mitigating computational and storage demands. Extensive experimental results validate the effectiveness of our approach, that enables real-time image editing capabilities on mobile devices and democratizes the power of diffusion models for efficient on-device computing.

**Limitation and Discussion.** Generating high-quality images using diffusion models can be challenging for diverse prompts, which in turn restricts the expansion of our training datasets. Moreover, utilizing diffusion models for data collection remains an expensive endeavor. Developing efficient techniques to rapidly construct well-paired and high-quality datasets from diffusion models would greatly enhance the training of E²GAN. Furthermore, while our approach does succeed in reducing training costs, alternative methods such as hyper-networks (Ha et al., 2017) have the potential to eliminate the need for training. This could result in the seamless obtaining real-time on-device GAN models as needed without any fine-tuning.

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

# A APPENDIX

## A.1 MORE IMPLEMENTATION DETAILS

**Details for Diffusion Model.** We apply most recent diffusion-based image editing models to create paired datasets, which include Stable Diffuison (SD) (Rombach et al., 2022), Instruct-Pix2Pix (IP2P) (Brooks et al., 2022), Null-text inversion (NI) (Mokady et al., 2022), ControlNet (Zhang & Agrawala, 2023), and Instruct Diffusion (Geng et al., 2023). For all these models, we use the checkpoints or pre-trained weights reported from their official websites[1].

More specifically, for SD, the strength, guidance scale, and denoising steps are set to $0.68$, $7.5$, and $50$, respectively. For IP2P, the images are generated with $100$ denoising steps using a text guidance of $7.5$ and an image guidance of $1.5$. For NI, each image is generated with $50$ denoising steps and the guidance scale is $7.5$. The fraction of steps to replace the self-attention maps is set in the range from $0.5$ to $0.8$ while the fraction to replace the cross-attention maps is $0.8$. The amplification value for words is $2$ or $5$, depending on the quality of the generation. For ControlNet, the control strength, normal background threshold, denoising steps, and guidance scale are $1$, $0.4$, $20$, and $9$, respectively. For Instruct Diffusion, the denoising steps, text guidance, and image guidance are set as $100$, $5.0$, and $1.25$, respectively.

**Hyperparameters in LoRA Rank Search.** During the process of searching LoRA rank, the rank $r_i$ for each crucial layer $i$ is upscaled once for every $e$ epochs until $r_i$ reaches the upper threshold $\tau_i$ for the layer $i$. In the experiments, $e$ is set as $10$. The rank threshold $\tau_i$ is determined by the size of the layer. More specifically, the crucial layers include: (1) four CONV-based upsampling layers with the shape as $[3, 64, 7, 7]$, $[64, 128, 3, 3]$, $[128, 256, 3, 3]$, and $[256, 256, 3, 3]$; (2) four corresponding downsampling layers by transpose CONV with the same set of weight shape as upsampling; and (3) transformer blocks with projection matrices $q$,$k$,$v$ with shape as $[256, 256]$, and multi-layer perceptron (MLP) module with shape as $[2048, 256]$ and $[256, 1024]$. Based on the weight size, the rank threshold $\tau$ is set as $1$, $4$, $16$, and $32$ for the four upsampling/downsampling layers, respectively, and $1$ for the layers in the transformer block. After the search process, the suitable rank is determined as $1$, $4$, $8$, $8$ for the four upsampling/downsampling layers.

## A.2 MORE ANALYSIS FOR THE EFFICIENT IMAGE-TO-IMAGE MODEL

**Effectiveness of Model Architecture.** Here we further show the effectiveness of our efficient model architecture design in complementary to the results in Sec. 3.2. We compare our 3RB+1TB design against the 9RB design used in pix2pix for several concept settings. The results are shown in Tab. 5 with both models trained on the entire training set of $800$ samples. From which we can see that the 3RB+1TB design can reach higher FID with fewer parameters and FLOPs (as in Tab. 1). For instance, a 3RB+1TB model in the target concept domain of `pale person` has a FID as $42.65$, decreasing the FID value by $6.49$ compared to the 9RB model of pix2pix.

Table 5: FID comparison between $E^2$GAN model architecture (3RB+1TB) and pix2pix (9RB) under the setting of training-from-scratch.

| Concept | $E^2$GAN (3RB+1TB) | Pix2pix (9RB) |
|---|---|---|
| Angry person | **49.56** | 55.16 |
| Pale person | **42.65** | 49.14 |
| Tan person | **42.47** | 51.37 |
| Young person | **51.27** | 56.10 |

**Sampling Operations for Transformer Block.** As mentioned in Sec. 3.2, we apply a downsampling operation with a CONV layer to halve the feature map size before sending it into the transformer

---

[1]SD v1.5: https://huggingface.co/runwayml/stable-diffusion-v1-5, IP2P: http://instruct-pix2pix.eecs.berkeley.edu/instruct-pix2pix-00-22000.ckp, NI: https://huggingface.co/CompVis/stable-diffusion-v1-4, ControlNet: https://huggingface.co/lllyasviel/ControlNet/blob/main/models/control_sd15_normal.pth, InstructDiffusion: https://github.com/cientgu/InstructDiffusion.

block, and use an upsampling layer implemented by transpose CONV operation to recover the feature map size for the following operations to reduce the amount of computations. We conduct another set of experiments on the Flicker Scenery dataset to see if these sampling operations can be replaced by pooling and unpooling operations, such that a smaller model size can be reached. We first train these two models on the selected prompts to get the base model. Then, we fine-tune the entire model with all the training data for a new concept. The comparision results are shown in Tab. 6. From the results, we can observe that though applying pooling operations can reduce the number of parameters from base model by 1.2M, the FID performance becomes much worse. Thus, we use CONV operation instead of pooling to tackle the feature map reduction and recovery for the transformer block.

Table 6: FID performance of replacing the downsampling and upsampling layers for the transformer block with Max Pool and Max Unpool operations.

| Operation | CONV + transpose CONV | Max Pool + Max Unpool |
|---|---|---|
| Model Size / Concept | 7.1M | 5.9M |
| Forest in the dark | **121.60** | 190.05 |
| Impressionism painting | **88.52** | 135.96 |
| Forest in the autumn | **88.82** | 141.29 |

### A.3 MORE ABLATION ANALYSIS FOR THE BASE MODEL

**Pre-train with Multiple Concepts for Conventional GAN Training.** We investigate if conventional GAN training such as pix2pix can benefit from fine-tuning a pre-trained base model, as leveraged in $E^2$GAN. To verify this, we follow the same step as $E^2$GAN to pre-train pix2pix with the selected 7 prompts/datasets on the Flicker Scenery dataset. Then, the base model is fine-tuned to adapt to other concepts. The results in Tab. 7 show that pix2pix does not gain much benefits from pre-training. Moreover, the performance becomes even worse, such as for the concept Vangogh style (FID degrades from 138.77 to 151.20 with a pre-trained base model). The results indicate that with our efficient architecture design, our base model possess the capability of a more general features and representations when trained on multiple concepts. The transformer block with self-attention modifies the image with a better holistic understanding and the cross-attention module takes the information from the given target concept. Thus, our method allows the new concept to better leverage existing knowledge, which is not possessed by prior methods.

Table 7: FID performance of fine-tuning from a pre-trained base model for pix2pix on Flicker Scenery dataset.

| Method | Pixpix | | $E^2$GAN |
|---|---|---|---|
| Pretrain / Concept | ✓ | ✗ | ✓ |
| Vangogh style | 151.2 | 138.77 | **117.41** |
| Add blossoms | 157.76 | 150.96 | **146.42** |
| Forest in the winter | 119.31 | 122.35 | **119.15** |

**Autoencoder as Pre-trained Base Model.** In $E^2$GAN, we first train the GAN model with multiple diverse concepts to get a pre-trained base model, and then fine-tune it to other concepts. We have shown multiple base model settings in Sec. 4.2. One may wonder if the pre-trained base model can be chosen as an auto-encoder, *e.g.,* the base model encodes the input data into a lower-dimensional representation and then decodes it back into the original data, instead of being trained on other concepts. To verify this, we conduct experiments by first training an auto-encoder on the original images in the subset of FFHQ (Karras et al., 2019) with only the $\ell_1$ loss in Eq. 1, then fine-tune the auto-encoder following the same method as fine-tuning a pre-trained GAN. The results are compared in Tab. 8. We find that auto-encoder is not comparable as fine-tuning a GAN trained on a single concept as old person, not to mention our base model that is pre-trained on multiple concepts. For instance, for the target style angry person, tuning from a base model pre-trained to generate old person can give an FID as low as 54.48, yet tuning from the auto-encoder results in a much

worse FID of 110.35. This might due to the simplicity of the auto-encoder, which only needs to generate the original image and does not necessarily include other semantic information, either coarse-grained global features, or fine-grained local details. In constrast, the GAN models include more information like texture or color, during training. From this observation, in E$^2$GAN, we adopt a model pre-trained on several concepts instead of using auto-encoder as base model.

Table 8: The FID performance of using autoencoder as the pre-trained base model.

| Concept / Base model | Angry person | White walker |
|---|---|---|
| Auto-encoder | 110.35 | 80.43 |
| Old person | 54.48 | 51.99 |
| **Ours** | **54.27** | **40.18** |

### A.4 ABLATION ON THE INFLUENCE OF LONGER TRAINING TIME

E$^2$GAN greatly saves the training time compared to conventional GAN training while maintaining good image synthesize ability. To see if training longer can lead to better performance, we add further experiments to increase the training time by doubling the training epochs. The results can be found in Tab. 9. The reported FID is evaluated on the model weights obtained at the end of training. The results show that training longer will not bring obvious performance improvements for E$^2$GAN, but leads to more computation cost. The results indicate that our efficient E$^2$GAN is able to reach good performance with fewer epochs compared to conventional GAN training.

Table 9: The FID comparison between training E$^2$GAN for 100 epochs and 200 epochs.

| Concept | Train 100 epoch | Train 200 epoch |
|---|---|---|
| Forest in the dark | 115.32 | 114.17 |
| Oil painting | **110.87** | 111.93 |
| Forest in the spring | **122.77** | 124.91 |

### A.5 ADDITIONAL QUALITATIVE RESULTS

We provide more example images generated by our approach and other baseline methods in Fig. 8, 9, 10, 11, and 12.

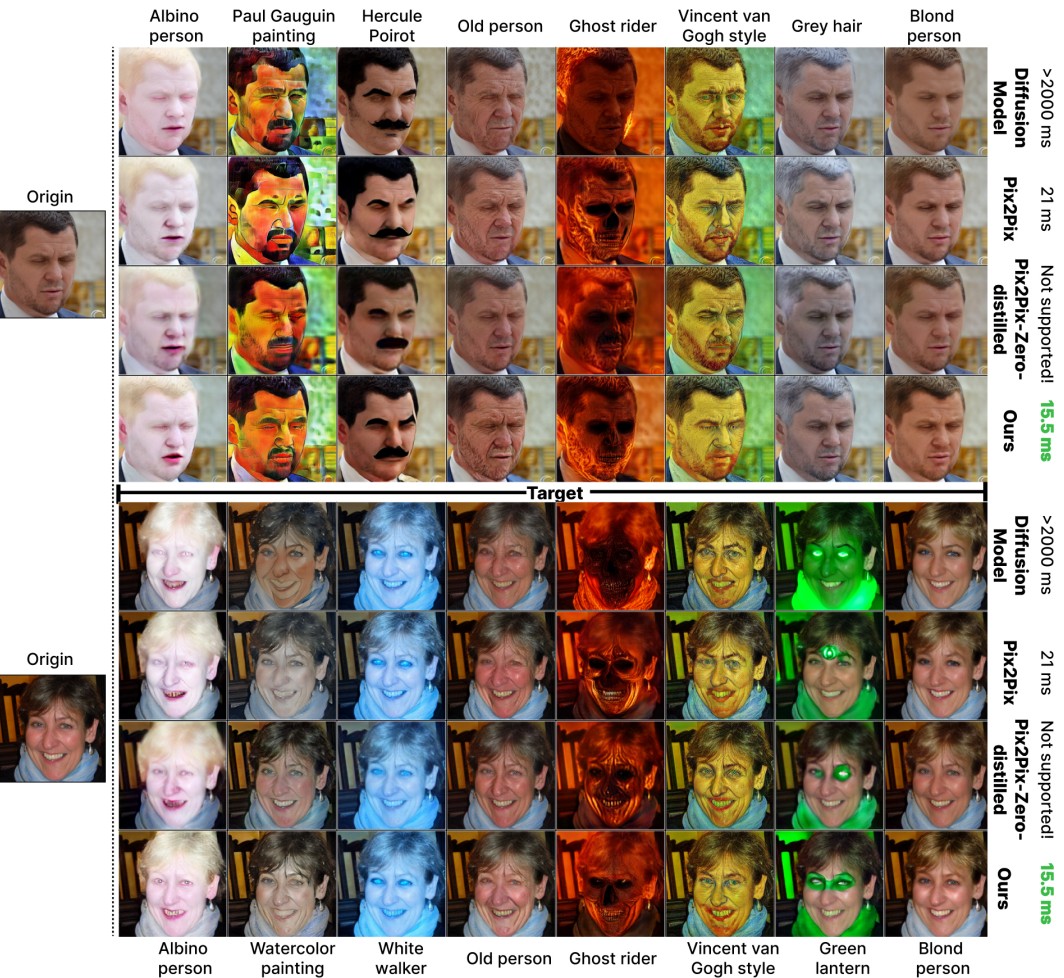

Figure 8: **Qualitative comparisons** on various tasks. The *leftmost* column shows two original images and the remaining columns present the corresponding synthesized images in the target concept domain, where target prompts are shown at the bottom row. We provide images generated by various models.

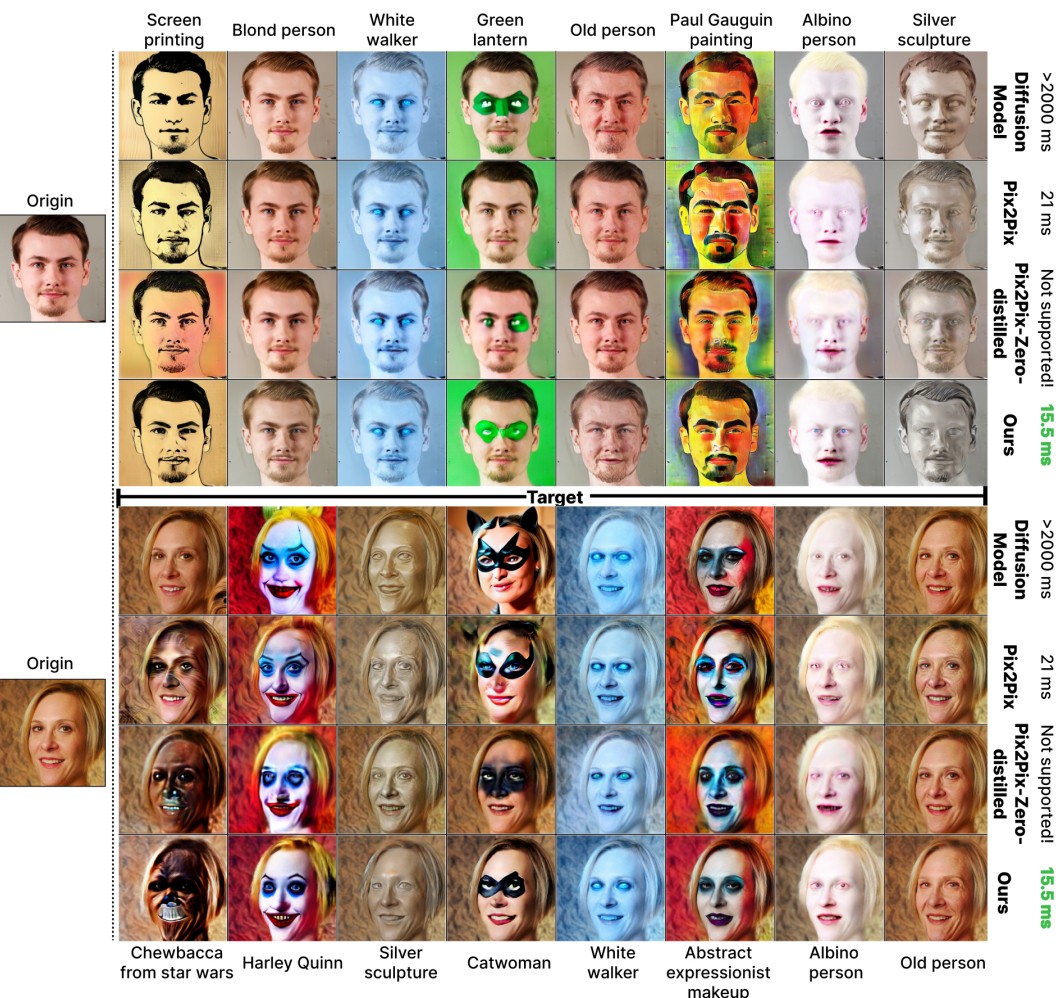

Figure 9: **Qualitative comparisons** on various tasks. The *leftmost* column shows two original images and the remaining columns present the corresponding synthesized images in the target concept domain, where target prompts are shown at the bottom row. We provide images generated by various models.

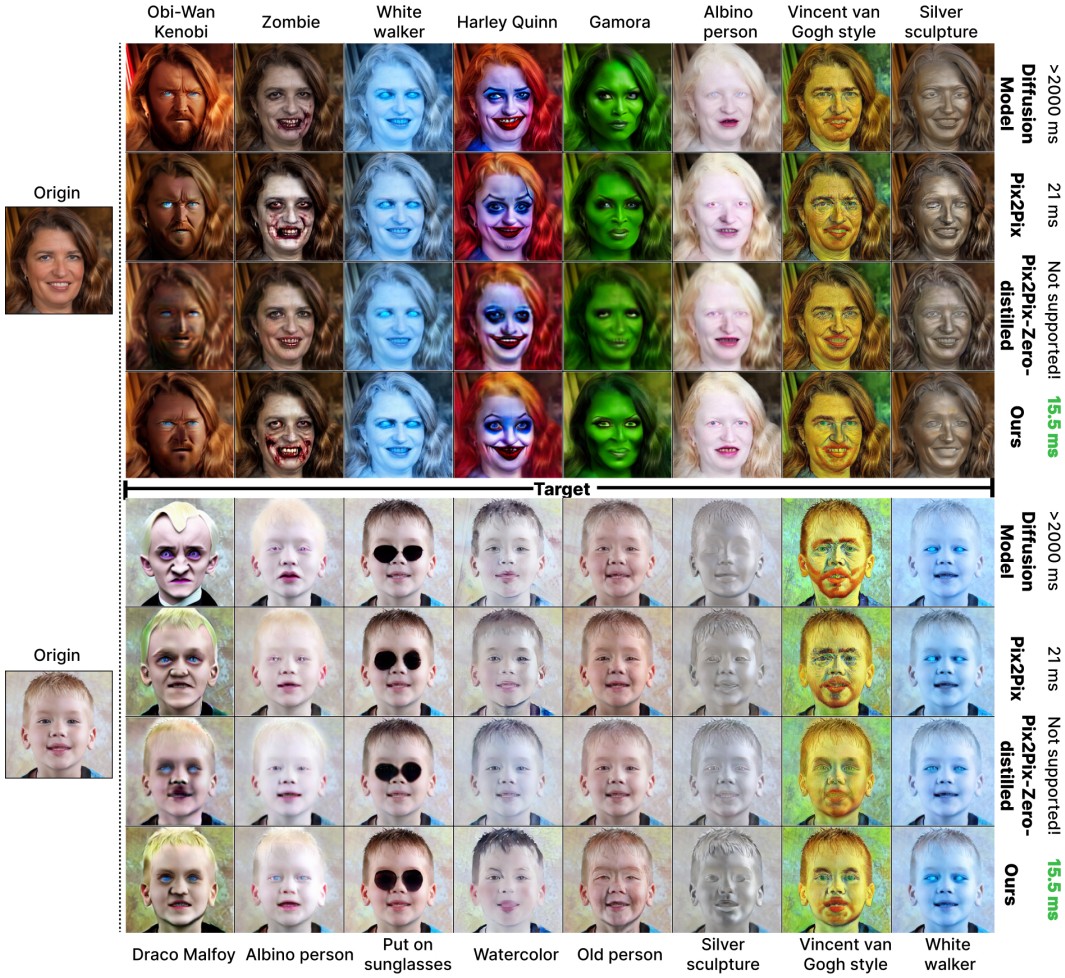

Figure 10: **Qualitative comparisons** on various tasks. The *leftmost* column shows two original images and the remaining columns present the corresponding synthesized images in the target concept domain, where target prompts are shown at the bottom row. We provide images generated by various models.

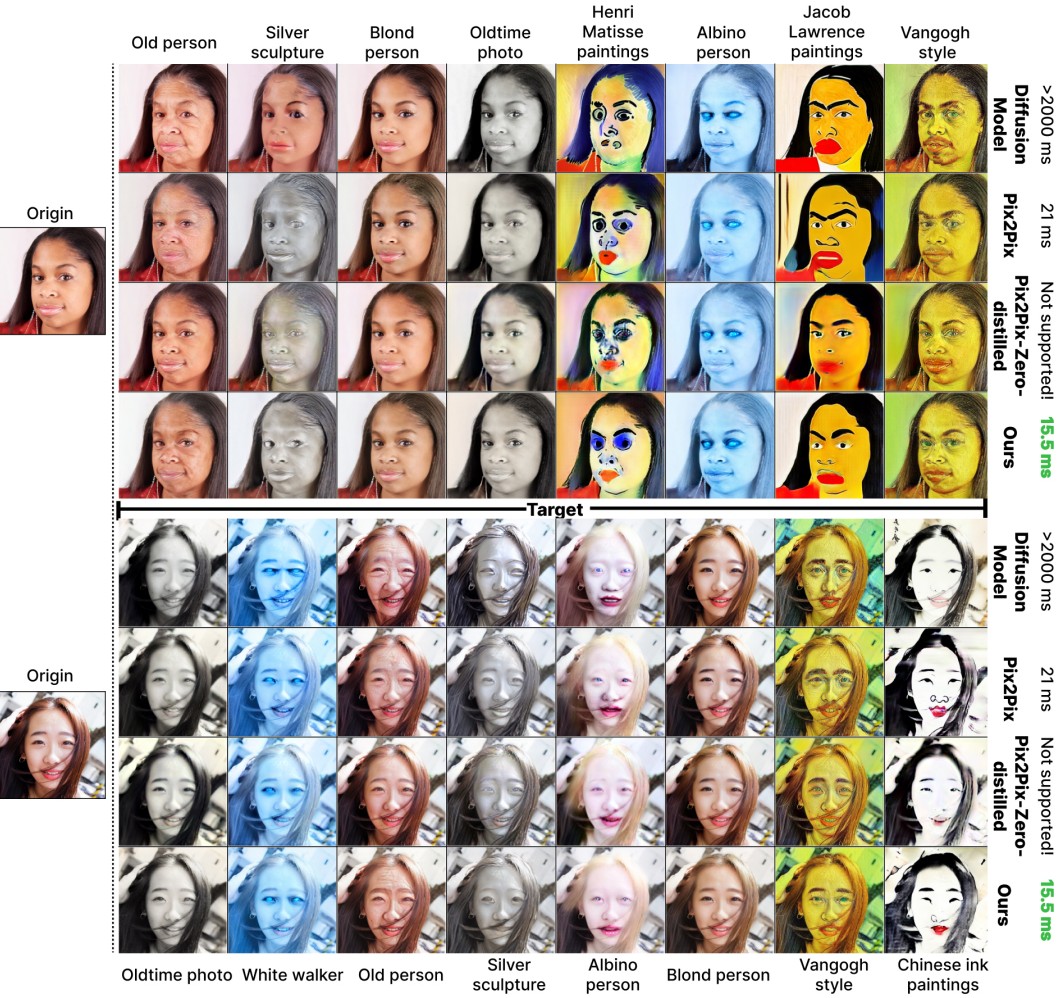

Figure 11: **Qualitative comparisons** on various tasks. The *leftmost* column shows two original images and the remaining columns present the corresponding synthesized images in the target concept domain, where target prompts are shown at the bottom row. We provide images generated by various models.

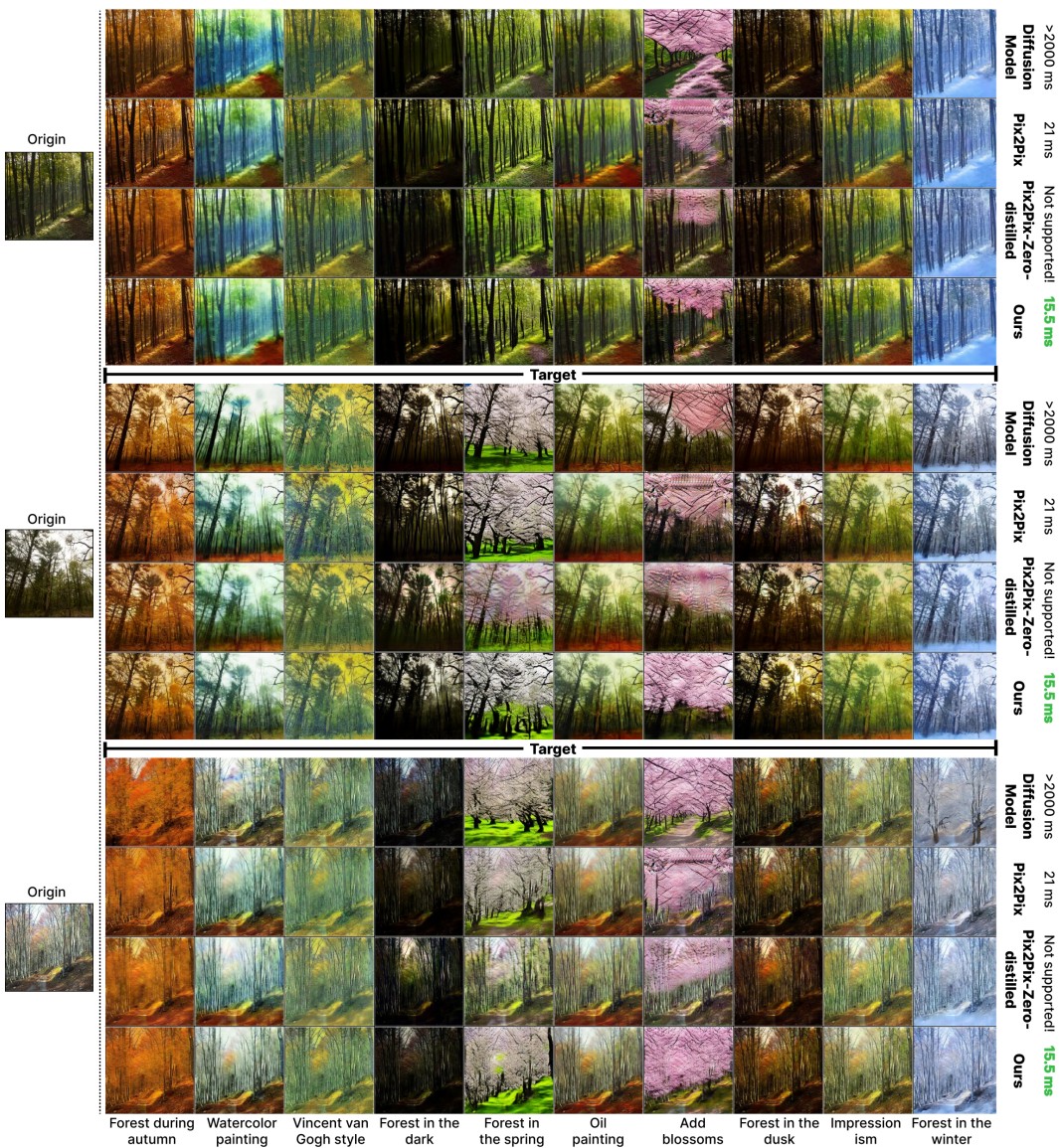

Figure 12: **Qualitative comparisons** on various tasks. The *leftmost* column shows two original images and the remaining columns present the corresponding synthesized images in the target concept domain, where target prompts are shown at the bottom row. We provide images generated by various models.

