# OpenReview forum: "E$^{2}$GAN: Efficient Training of Efficient GANs for Image-to-Image Translation"
_ICLR.cc/2024/Conference — ICLR 2024 Conference Withdrawn Submission_

### Official Review · Reviewer_iBPB · 2023-10-18

**Soundness:** 3 good
**Presentation:** 3 good
**Contribution:** 2 fair
**Rating:** 5
**Confidence:** 3

**Summary:**

This paper focuses on improving the inference and training efficiency of GANs for image-to-image translation. For inference efficiency, the authors design an efficient encoder-decoder backbone with fewer convolutional blocks and one additional transformer block. For training efficiency, the authors first pre-train a text-conditioned GAN on various concepts, then employ LoRA to minimize the number of trainable parameters, and finally leverage feature clustering to reduce the size of the training dataset. The proposed framework delivers good empirical performance, in accuracy, inference efficiency and training efficiency.

**Strengths:**

The paper is well-written and easy to follow. The authors offer a comprehensive introduction to the knowledge transfer pipeline for GANs in Section 3.1. This provides clarity for readers unfamiliar with the domain. The proposed framework is technically sound and delivers good empirical results. Furthermore, the provided implementation details are extensive, enabling researchers and practitioners in the field to replicate the findings and expand on this method.

**Weaknesses:**

The proposed framework appears to be "a bag of tricks":
1. Improved efficient backbone with hybrid convolution-attention architecture.
2. Base promptable image translation model.
3. Parameter-efficient fine-tuning using LoRA.
4. Training data subsampling by clustering CLIP features.

Diving deeper into these components, (1) is not a novel contribution. A hybrid convolution-attention architecture is already a common practice in the vision community and has been extensively explored in many papers, such as MobileViT [Mehta et al., ICLR 2022] and EfficientViT [Han et al., ICCV 2023]. Likewise, LoRA in (3) is widely used for parameter-efficient fine-tuning (PEFT), spanning both vision and language communities. While (2) and (4) are not conceptually new, it is still a good engineering effort to explore them in the context of GANs. Overall, it is unclear to me whether the technical contributions presented in this paper meet the publication standards of ICLR.

Further compounding this is the observation that these four techniques appear to operate in silos, without much interdependence. Despite the framework's impressive end-to-end speedup, the individual impact of each component remains unclear. A comprehensive breakdown is crucial to elucidate the distinct contributions and effectiveness of each component to readers.

**Questions:**

My primary concerns are outlined in the weaknesses section. Beyond these points, I have several additional questions/comments:
* Table 1 only presents efficiency metrics for different models. Please kindly add accuracy metrics for image generation quality as well.
* The term "data distillation" may be misleading due to its resemblance to "dataset distillation". It might be more appropriate to use "distillation" or "model distillation".

---

> ### Author Response · Authors · 2023-11-17
> **Reply by authors**
>
> **We thank the reviewer for recognizing our work as well-written, technically sound, with extensive implementation details. We would like to address the questions as below.**
>
> ***
>
> **Q1: The individual impact of each component.**
>
> We agree with the reviewer that showing the individual impact of each component is important, and we did study the influence of each component in the ablation studies. More specifically, The study of the model architecture is in Fig.2 in the main paper and A.2 in the appendix. The study of the base model is shown in Table 3 in the main paper and A.3 in the appendix. The influence of parameter-efficient fine-tuning using LoRA is shown in Table 4 while the training data subsampling is shown in Fig.7.
>
> ***
>
> **Q2: About accuracy metrics for Table 1.**
>
> We thank the reviewer for the suggestion. The accuracy metrics are mainly shown in Table 2 as it is associated with the dataset and target concepts.
>
> ***
>
> **Q3: The term "data distillation" may be misleading due to its resemblance to "dataset distillation". It might be more appropriate to use "distillation" or "model distillation".**
>
> We thank the reviewer for the suggestion. We would revise our manuscript with this suggestion.

---

### Official Review · Reviewer_Jjyt · 2023-10-29

**Soundness:** 2 fair
**Presentation:** 2 fair
**Contribution:** 1 poor
**Rating:** 3
**Confidence:** 3

**Summary:**

This work proposes a framework to distill features from a pre-trained large diffusion-based image-to-image model into a GAN-based image-to-image model for fast on-device inference performance. The authors replace conv with attention module in the img2img GAN to achieve more efficient inference, and they also inject text condition into the GAN framework. Lastly, they empirically show such a design does not need to retrain for a new kind of edit, which greatly reduces storage and training efforts. Overall the paper is easy to follow and well-motivated.

**Strengths:**

– The author conducts extensive ablation study on the design choices of the architecture in terms of the number of different blocks
– Table 1 shows strong inference speed improvement in terms of both model size and FLOPs, comparing to baselines

**Weaknesses:**

– The author only conducts experiments on small-scale datasets, i.e. FFHQ and Flickr-Scenery with 1000 images. Also, the resolution is only 256, so it is hard to verify the claim that the base GAN can be trained on various text condition prompts on a large scale.

– Lack of technical novelty. The proposed base GAN architecture and also how to do text conditioning is very similar to GigaGAN.

**Questions:**

– I feel it lacks comparison to recent proposed efficient diffusion-based image2image models
– Is it practical to use the proposed method considering significant quality drops after distillation?

---

> ### Author Response · Authors · 2023-11-17
> **Reply by authors**
>
> **We thank the reviewer for acknowledging our extensive experiments and performance improvement in terms of training cost and inference speed.**
>
> ***
>
> **Q1: About the experimental datasets.**
>
> We would like to emphasize that the main contribution of our paper is to achieve efficient GAN training with the knowledge from diffusion models. Scale is not our major concern. And the fewer training data is used, the more efficient the approach is. Our contribution is not solely to provide a base model worked on various text condition, but it is to leverage the base model for fast and efficient fine-tuning to new concepts.
>
> ***
>
> **Q2: About technical novelty.**
>
> GigaGAN focuses on scale up GAN models to achieve text-to-image synthesis. It contains 1 billions parameters and the latency is far beyond real-time. Differently, our work considers to efficiently train efficient GAN models that achieve real-time inference on mobile device. From the model architecture perspective, our model design is more efficient. Besides, Our main contribution is to build an efficient distillation pipeline (use fewer training efforts)  that brings the image editing capability of diffusion models to an efficient GAN (achieves real-time inference on mobile), which is very different from GigaGAN. More specifically, our contributions include:
> - We introduce a text-conditional base GAN model incorporating CONV layers with attention mechanisms. With this design, the base model can be trained on various prompts and the corresponding edited images obtained from diffusion models. Based on the base GAN model, a new editing concept only requires a fine-tuning process with reduced training costs compared to train-from-scratch setting.
> - We identify partial crucial layers of the base GAN model, and further apply LoRA on the crucial layers with a simple and effective rank search process, instead of fine-tuning the entire base model. With this approach, we can greatly save the training and storage cost for a new concept.
> - We reduce the amount of training data used to obtain a new concept based on similarity clustering.
>
> ***
> **Q3:  About comparisons to efficient diffusion-based image2image models**
>
> As mentioned in Sec. 1 and Tab. 1, efficient diffusion model designs still struggle to obtain models that can run in real-time on mobile devices. Yet our work focuses on real-time image editing on mobile. For instance, the latest SnapFusion takes nearly 2 seconds to inference on mobile device.
>
> ***
>
> **Q4: About practical usage.**
>
> We did not observe significant quality drops after diffusion. The computed FID score is calculated by viewing the images generated by diffusion models as grountruth. It does not mean that diffusion model generates better images. For instance, as in Fig. 5, for the target concept “green lantern”, our method generates more meaning images.

---

### Official Review · Reviewer_myfn · 2023-11-01

**Soundness:** 2 fair
**Presentation:** 3 good
**Contribution:** 2 fair
**Rating:** 5
**Confidence:** 4

**Summary:**

This work is conducted towards efficient training and efficient inference for image-to-image translation via generative adversarial network (GAN). The paper constructs the base GAN model with a text conditional image generation task trained with various prompts and corresponded edited images from diffusion models, and then the pre-trained GAN model is fine-tuned with the analysis of effective partial weights for reducing the training cost and saving the storage.

**Strengths:**

- The efficient image editing, addressed by this work, is useful and helpful for the current mobile devices.
- It seems that the final model is light-weight and can be run fast on mobile devices.

**Weaknesses:**

- The novelty of this work is confusing. This work combines many existing techniques to pre-train and fine-tune a light-weight GAN model for efficient image-to-image translation, however, it seems that, not only the network architecture and loss functions, but also the transfer learning and knowledge transfer strategies, are not new and it is confusing what are the key contributions of this work that can differ from previous techniques and also are meaningful to the related study. Besides, the whole training and inference pipeline is also not clear enough, for example, how and why does this work distill GANs from diffusion models significantly efficient? Although this is the key question raised by this work, it is not easy to get the satisfactory answer after reading through the paper.
- The evaluation experiments are insufficient. The evaluation is mainly conducted on the FFHQ and Flickr-Scenery datasets compared with pix2pix and pix2pix-zero-distilled in terms of FID, and if it is towards the mobile devices, except for the efficient requirement, it is also important for the acceptable results required by the users, but it seems that this paper concerns not more about this factor and provides not enough evidence on the requirement.

**Questions:**

- What are the key contributions of this work that can differ from previous techniques and also are meaningful to the related study?
- What are the performance towards the real-world applications on mobile devices required by the users?

---

> ### Author Response · Authors · 2023-11-17
> **Reply by authors**
>
> **We greatly appreciate the reviewer for recognizing our efficient image editing for mobile devices are useful and helpful.  And thank you very much for the very constructive comments, which are responded below.**
>
> ***
>
> **Q1: The novelty and the key contributions of this work.**
> We thank the reviewer for the comments and will revise the writing to make the contributions more clear. Our main contribution is to build an efficient distillation pipeline (use fewer training efforts)  that brings the image editing capability of diffusion models to an efficient GAN (achieves real-time inference on mobile), which is a new research direction. Different from prior work considers data distillation to transfer knowledge from diffusion models to GANs for efficient inference, our method not only achieves efficient inference, but also efficient training to obtain GANs . More specifically, our contributions include:
>
> - We introduce a text-conditional base GAN model incorporating CONV layers with attention mechanisms. With this design, the base model can be trained on various prompts and the corresponding edited images obtained from diffusion models. Based on the base GAN model, a new editing concept only requires a fine-tuning process with reduced training costs compared to train-from-scratch setting.
> - We identify partial crucial layers of the base GAN model, and further apply LoRA on the crucial layers with a simple and effective rank search process, instead of fine-tuning the entire base model. With this approach, we can greatly save the training and storage cost for a new concept.
> - We reduce the amount of training data used to obtain a new concept based on similarity clustering.
>
> ***
>
> **Q2: About the evaluation experiments are insufficient.**
>
> For mobile applications, we evaluate our proposed method from two fundamental aspects, i.e., image quality (in terms of FID)  and latency. We appreciate your emphasis on user acceptability as a crucial factor for mobile device applications. We will incorporate user-centric metrics such as human evaluation results.

---

### Official Review · Reviewer_DSwv · 2023-11-02

**Soundness:** 2 fair
**Presentation:** 3 good
**Contribution:** 2 fair
**Rating:** 3
**Confidence:** 4

**Summary:**

This paper provides an approach to enable flexible real-time image editing on mobile devices. It use Stable Diffusion to generate paired data to train Pix2pix. This approach overcomes the high computational requirements of traditional image editing with diffusion models. The research introduces more efficient methods for distilling GANs from diffusion models.  The proposed techniques include: 1) developing an attention-based network architecture for image-to-image translation on mobile devices,  2) implementing a hybrid training pipeline to adapt a pre-trained text-conditioned GAN model to different concepts efficiently,  and 3) investigating the minimum amount of data needed to train each GAN, further reducing the training time.

**Strengths:**

1) This paper build a connection between Stable Diffusion and GANs. It makes sense to use SD to create paired data, since there appears effective techniques to do image editing with SD.

2) This paper is easy to follow.

3) Authors conduct extensive experiments that demonstrate the proposed techniques.

**Weaknesses:**

I have some questions about this paper.

1) This proposed method are not supervising, since it is not easy to collect paired image from SD. In fact it is not hard to get this idea to create data.  I think this idea has less contribution for I2I translation community.

2) This paper combines more techniques into one paper. I see it is useful, but stacking a few techniques into one paper make me feel hard to  grasp the key contribution.  I could not grasp the important contribution from this paper.

3) Also transferring knowledge from SD to Pix2pix model is not much interesting.   Is the approach  generalizable for other tasks?

4) The shown figures are not too much interesting. Comparing the powerful editing ability of SD, using Pix2pix to do this task is not attracting.

5) Using attention mechanism  I think is not contribution, which just follows what have done in SD.

**Questions:**

1) Since this paper combines a few contributions into one paper,  I could not grasp the key contribition.

2) This paper is  less interesting.

---

> ### Author Response · Authors · 2023-11-17
> **Reply by authors**
>
> **We thank the reviewer for acknowledging our paper building connections between diffusion models and GANs, easy to follow, and containing extensive experiments.**
>
> ***
>
> **Q1: Not easy to collect paired image from SD.**
>
> Paired data collection is not a hard issue. As mentioned in Sec. 3.1, diffusion models are used to generate edited images in the target domain, and the edited images and original images (e.g., images in the FFHQ dataset) naturally form paired data for GAN training.
>
> ***
>
> **Q2: About contribution to I2I community.**
>
> As mentioned in Sec. 1 and Sec. 2, as diffusion models are too bulky and contain multiple sampling steps during inference, making them achieve real-time on-device inference is hard. Our paper mitigates this problem by bringing the image editing capability of diffusion models to lightweight and real-time GANs with distillation. To make the distillation process more efficient, we introduce a hybrid training pipeline that efficiently adapts a pre-trained text-conditioned GAN model to different concepts while substantially reducing computational and storage costs. We also reduce the amount of  training data necessary to train each GAN.
>
> ***
>
> **Q3: About key contributions of the paper.**
>
> Our main contribution is to build an efficient distillation pipeline (use fewer training efforts)  that brings the image editing capability of diffusion models to an efficient GAN (achieves real-time inference on mobile). More specifically, our contributions include:
> - We introduce a text-conditional base GAN model incorporating CONV layers with attention mechanisms. With this design, the base model can be trained on various prompts and the corresponding edited images obtained from diffusion models. Based on the base GAN model, a new editing concept only requires a fine-tuning process with reduced training costs compared to train-from-scratch setting.
> - We identify partial crucial layers of the base GAN model, and further apply LoRA on the crucial layers with a simple and effective rank search process, instead of fine-tuning the entire base model. With this approach, we can greatly save the training and storage cost for a new concept.
> - We reduce the amount of training data used to obtain a new concept based on similarity clustering.
>
> ***
>
> **Q4: About the task of transferring knowledge from SD to Pix2pix model. Is the approach generalizable for other tasks?**
>
> Transferring knowledge from diffusion models to GANs has important advantages. Diffusion models are large in size and have high computation costs. Though recent works attempt to accelerate the image generation process of the diffusion models, the results are still far beyond real-time on-device inference, which is important for user experience. Our knowledge transferring pipeline can obtain GANs for new concepts in an efficient way, and allow real-time inference after deployment, which takes advantage of diffusion models but avoids the high storage cost and long latency issue.
>
> ***
>
> **Q5: About using attention mechanism.**
>
> We acknowledge that attention mechanisms have been utilized in diffusion models, yet using it in GAN models is not that straightforward. Considering our objective is to achieve real-time inference on mobile while maintaining good image quality, the number and position of transformer blocks are important aspects to be investigated, as in Sec. 3.2.